# Human Bone Marrow Mesenchymal Stem Cells Promote the M2 Phenotype in Macrophages Derived from STEMI Patients

**DOI:** 10.3390/ijms242216257

**Published:** 2023-11-13

**Authors:** Víctor Adrián Cortés-Morales, Wendy Guadalupe Vázquez-González, Juan José Montesinos, Luis Moreno-Ruíz, Selene Salgado-Pastor, Pamela Michelle Salinas-Arreola, Karla Díaz-Duarte, Adriana Karina Chávez-Rueda, Luis Chávez-Sánchez

**Affiliations:** 1Unidad de Investigación Médica en Inmunoquímica, Hospital de Especialidades, Centro Médico Nacional Siglo XXI, Instituto Mexicano del Seguro Social, Mexico City 06720, Mexico; v.adrian.cortes@gmail.com; 2Unidad de Investigación Médica en Enfermedades Metabólicas del Hospital de Cardiología, Centro Médico Nacional Siglo XXI, Instituto Mexicano del Seguro Social, Mexico City 06720, Mexico; wendisya@live.com.mx (W.G.V.-G.);; 3Unidad de Investigación Médica en Enfermedades Oncológicas, Hospital de Oncología, Centro Médico Nacional Siglo XXI, Instituto Mexicano del Seguro Social, Mexico City 06720, Mexico; 4División de Cardiología del Hospital de Cardiología, Centro Médico Nacional Siglo XXI, Instituto Mexicano del Seguro Social, Mexico City 06720, Mexico; 5Unidad de Investigación Médica en Inmunología, Hospital de Pediatría, Centro Médico Nacional Siglo XXI, Instituto Mexicano del Seguro Social, Mexico City 06720, Mexico

**Keywords:** acute ST-elevation myocardial infarction, bone marrow mesenchymal stem/stromal cells, M1 macrophages, M2 macrophages, regulatory T-cells

## Abstract

Acute ST-elevation myocardial infarction (STEMI) leads to myocardial injury or necrosis, and M1 macrophages play an important role in the inflammatory response. Bone marrow mesenchymal stem/stromal cells (BM-MSCs) are capable of modulating macrophage plasticity, principally due to their immunoregulatory capacity. In the present study, we analyzed the capacity of MSCs to modulate macrophages derived from monocytes from patients with STEMI. We analyzed the circulating levels of cytokines associated with M1 and M2 macrophages in patients with STEMI, and the levels of cytokines associated with M1 macrophages were significantly higher in patients with STEMI than in controls. BM-MSCs facilitate the generation of M1 and M2 macrophages. M1 macrophages cocultured with MSCs did not have decreased M1 marker expression, but these macrophages had an increased expression of markers of the M2 macrophage phenotype (CD14, CD163 and CD206) and IL-10 and IL-1Ra signaling-induced regulatory T cells (Tregs). M2 macrophages from patients with STEMI had an increased expression of M2 phenotypic markers in coculture with BM-MSCs, as well as an increased secretion of anti-inflammatory cytokines and an increased generation of Tregs. The findings in this study indicate that BM-MSCs have the ability to modulate the M1 macrophage response, which could improve cardiac tissue damage in patients with STEMI.

## 1. Introduction

Atherosclerosis is the most common underlying cause of cardiovascular disease, and ischemic heart disease is one of the leading causes of death worldwide [1]. Acute coronary syndromes (ACSs) comprise the acute manifestations of coronary artery disease, including unstable angina, non-ST-segment elevation myocardial infarction and ST-segment elevation myocardial infarction (STEMI). STEMI results from the total occlusion of an epicardial coronary artery caused by a thrombus (blood clot) that develops via the destabilization and rupture of a coronary atherosclerotic plaque [2]. This blockage causes cardiac tissue necrosis or the death of a portion of the cardiac muscle, which induces the release of the intracellular contents of the cardiomyocytes; in addition, the damaged extracellular matrix releases endogenous warning signals, which cause an intense inflammatory reaction [3,4].

Although the inflammatory response in myocardial tissue is essential for the restoration of structure and function, an uncontrolled or unresolved inflammatory process can trigger tissue damage [5,6]. In this regard, macrophages have essential functions in inflammation and tissue repair processes [6,7]. The microenvironment and released signals induce macrophage polarization toward an inflammatory (M1) or an anti-inflammatory (M2) phenotype [8]. M1 macrophages dominate in the early stages after myocardial infarction (MI), whereas M2 macrophages are the major cells that exist post MI in the mouse heart [9]. M1 macrophages secrete inflammatory cytokines, chemokines and matrix metalloproteinases (MMPs) to help clear cell debris and degrade the extracellular matrix [10,11]. The prolonged presence of M1 macrophages can lead to the expansion of the infarct size and can impede the resolution of inflammation and scar formation [12]. In contrast, anti-inflammatory M2 macrophages are reparative. These macrophages produce anti-inflammatory cytokines and proangiogenic and pro-reparative factors, and engulf apoptotic cells to facilitate neoangiogenesis and scar repair [11].

The immunoregulation of the balance between M1 and M2 macrophages could improve myocardial repair and function following MI [6,7]. Given that mesenchymal stem cells (MSCs) might regulate the inflammatory immune response, the transplantation of bone marrow MSCs (BM-MSCs) into the infarcted area of the heart is an interesting approach to regenerating the myocardium. In this regard, animal mouse models of myocarditis have shown that the administration of MSCs reduces the severity of myocarditis and decreases the number of proinflammatory monocytes [13]. Similarly, BM-MSC transplantation has been shown to induce cardiac tissue regeneration following acute myocardial infarction in mice [14]. A randomized controlled trial of patients with STEMI evaluated its efficacy and safety, and showed that BM-MSC treatment can significantly improve the left ventricular ejection fraction (LVEF) [15]. Although these trials clearly show the efficacy of MSCs for the treatment of myocardial infarction, the mechanisms by which MSCs improve cardiac functional activity are not entirely clear.

MSCs promote the survival of monocytes derived from healthy subjects, induce polarization toward M2 and enhance CD209, CD206 and CD163 expression, as well as IL-10 production in these macrophages [16,17,18]; they also and promote the preferential expansion of T cells with regulatory function (Tregs) [18,19], which inhibit the proliferation of CD8^+^ T cells [16,17]. These data suggest that MSCs may play an important role in alternatively regulating macrophage induction. However, our understanding of MSC and macrophage interactions in STEMI remains incomplete. The aim of the present study was to evaluate the effect of BM-MSCs on the immunophenotype characteristics of monocyte-derived macrophages from patients with STEMI. We showed that BM-MSCs induce the generation of M2-polarized macrophages.

## 2. Results

### 2.1. Population Characteristics

The characteristics of the study population are shown in Table 1. The STEMI patients included 55 men and 4 women with a mean age of 66 ± 12 years. Of these patients, 27 had diabetes, 37 had systemic arterial hypertension, 33 were smokers, 28 had obesity and 37 had hyperlipidemia. Twenty healthy subjects were included; these were 9 men and 11 women with an average age of 34 ± 11 years, without cardiovascular risk factors.

### 2.2. Cytokine Profile Associated with M1 and M2 Macrophages in Patients with STEMI

Inflammatory and anti-inflammatory cytokine secretion is particularly active in cardiac conditions [20]. We initially determined the circulating cytokines associated with M1 and M2 macrophages in patients with STEMI. We found that STEMI patients were characterized by higher levels of inflammatory cytokines than healthy subjects: IL-1β (STEMI: 12.41 ± 8.74 pg/mL; healthy: 7.08 ± 6.05 pg/mL; *p* < 0.05), IL-6 (STEMI: 5.78 ± 9.47 pg/mL; healthy: 1.2 ± 0.00 pg/mL; *p* < 0.05), IL-12p70 (STEMI: 5.42 ± 2.63 pg/mL; healthy: 3.78 ± 2.36 pg/mL; *p* < 0.05) and IP-10 (STEMI: 79.35 ± 30.51 pg/mL; healthy: 60.82 ± 32.52 pg/mL; *p* < 0.05), as shown in Figure 1A. Surprisingly, we did not find changes in the concentration of molecules that favor M2 polarization (Figure 1B). These findings indicate that patients with STEMI have a high plasma concentration of M1 macrophage-associated inflammatory cytokines, suggesting that M1 macrophages influence this response in patients with STEMI.

### 2.3. BM-MSCs Promote Macrophage Differentiation

BM-MSCs regulate the inflammatory microenvironment and influence cell differentiation, maturation and proliferation [21,22]. In this sense, we evaluated whether BM-MSCs modified the differentiation phenotypes of macrophages through the marker CD68 [23]. In our results, we found that in the presence of media induced by M1 or M2 macrophages, the expression of CD68 on macrophages increased in the coculture of BM-MSCs in patients with STEMI (fold change: M1: 1.34 ± 0.18, M2: 1.57 ± 0.33; *p* < 0.05), an effect not determined on macrophages in healthy donors. However, M0 macrophages from healthy donors increased the expression of CD68 compared to macrophages from patients (M0: 1.73 ± 0.31; *p* < 0.05). (Figure 2). This result indicates that the presence of BM-MSCs favors the maturation of monocytes derived from patients with STEMI toward macrophages, dependent on a microenvironment toward M1 or M2 macrophages.

### 2.4. BM-MSCs Drive Macrophages into an M2 Macrophage Phenotype on Macrophages Derived from Patients with STEMI

The administration of BM-MSCs in humans or animal models decreases cardiac affectations [24], and these cells modify the expression of phenotypic markers on M1 and M2 macrophages [22]. In our work, when evaluating the expression of M1 macrophage molecules, we found that HLA-DR (*p* < 0.05) and CD80 (*p* < 0.05) levels were increased and that this expression was maintained under M0- and M1-inducing conditions in macrophages derived from patients with STEMI in the presence of BM-MSCs, unlike macrophages derived from healthy donors, which expressed lower levels of these markers (Figure 3A). Only in the presence of M1-inducing medium did the presence of BM-MSCs decrease the expression of CD86 in M1 macrophages derived from STEMI patients (fold change: 0.70 ± 0.22; *p* < 0.05) compared to those of the controls (Figure 3A). Meanwhile, the M2 macrophage-inducing medium induced HLA-DR (fold change: 1.11 ± 0.28; *p* < 0.05) expression in macrophages from STEMI patients relative to healthy subjects (Figure 3A). These data suggest that BM-MSCs do not downregulate the expression of M1 markers in macrophages derived from STEMI patients. Conversely, we evaluated the expression of M2 markers in the different induction media on macrophages from patients. Interestingly, we determined that culture with BM-MSCs increased the expression of M2 markers in any condition of macrophage-inducing medium from patients with STEMI: CD14 (fold change: M0: 2.29 ± 0.58; M1: 1.84 ± 1.16, M2: 1.73 ± 0.42; *p* < 0.05), CD163 (fold change: M0: 1.47 ± 0.31; M1: 1.66 ± 0.26, M2: 1.73 ± 0.73; *p* < 0.05) and CD206 (fold change: M0: 1.29 ± 0.15; M1: 1.45 ± 0.11, M2: 1.84 ± 2.73; *p* < 0.05), similar to healthy donor-derived macrophages, compared to macrophages in the absence of BM-MSCs (Figure 3B). These results suggest that BM-MSCs increased M2 macrophage markers in both M1 and M2 conditions in macrophages from patients with STEMI.

### 2.5. BM-MCS Coculture Increases Levels of Anti-Inflammatory Cytokines

Previous studies have indicated that the interaction of macrophages with BM-MSCs increases the production of anti-inflammatory cytokines and decreases inflammation [25]. Next, we investigated the effect of BM-MCSs in coculture with macrophage induction medium on the production of cytokines. Consistent with our previous findings, we observed the expression of M1 markers in macrophages cocultured with BM-MSCs. IL-10 and IL-1Ra levels were analyzed in cocultures of BM-MCSs with macrophages. In the presence of BM-MSCs, we found an increase in cytokines characteristic of M2 polarization, such as IL-10 (M0: 5.96 ± 1.78 pg/mL; M1: 44.81 ± 27.83 pg/mL; *p* < 0.05) and IL-1Ra (M0: 13.97 ± 6.92 pg/mL; M1: 107.07 ± 136.95 pg/mL; M2: 109.45 ± 103.53 pg/mL; *p* < 0.05), in the induction medium evaluated compared to macrophages cultured in the absence of BM-MCSs (Figure 4). These results indicate that the presence of BM-MSCs favors the secretion of cytokines characteristic of M2 polarization in STEMI-derived macrophages.

### 2.6. Effect of Macrophage Polarization on BM-MSCs

Several reports have indicated that MSCs express/secrete molecules that drive M2 polarization in macrophages [26]. We evaluated the IL-10, M-CSF and CD54 molecules, which have been reported to lead to anti-inflammatory polarization in macrophages and that have been shown to be expressed in BM-MSCs. Cocultures of BM-MSCs with macrophages derived from patients with STEMI showed decreased IL-10 expression in BM-MSCs (M0: 33.4% ± 11.29; *p* < 0.05), M-CSF (M0: 70.42% ± 7.53; M2: 57.7% ± 13.53; *p* < 0.05) and CD54 (M0: 6.82% ± 1.93; *p* < 0.05) compared to BM-MSCs cocultured with healthy donor-derived macrophages. Conversely, BM-MSCs showed an increase in CD54 expression under M1 macrophage-inducing conditions (Figure 5). These data suggest that BM-MSCs maintain the expression of IL-10 and M-CSF, and increase CD54 levels in the presence of M1 macrophages, which could favor the transition to M2 macrophages. Meanwhile, BM-MSCs favor M2 macrophages through IL-10.

### 2.7. BM-MSCs Coculture Increases the Generation of Regulatory T Cells

Tregs, especially the subset of CD4^+^Foxp3^+^ Tregs, are critical in maintaining immune homeostasis and regulating inflammation in myocardial infarction [27]. We evaluated the effect of macrophages cocultured with BM-MSCs on the induction of Tregs. Therefore, BM-MSCs were cocultured without contact with macrophages under different macrophage-polarizing conditions; then, macrophages were cocultured with Tregs. We found an increase in the generation of Treg cells in the absence of induction medium in the M0 condition (STEMI 17.57% ± 9.18; STEMI/BM-MSCs: 30.2% ± 4.45; *p* < 0.05) and in the presence of induction medium in the M2 condition (STEMI: 19.80% ± 8.35; STEMI/BM-MSCs: 32.65% ± 2.52; *p* < 0.05) (Figure 6). Concurrently, the condition of the M1 inductor medium does not significantly increase the number of Tregs. In healthy subjects, we found that any macrophage-polarizing condition with BM-MSCs induces the generation of Tregs (Figure 6). These data suggest that BM-MSCs cocultures with M0, MI and M2 macrophages induce Treg generation in a similar manner in patients and healthy donors; this generation of Tregs can be influenced by the cytokines IL-10 and IL-Ra (Figure 6).

## 3. Discussion

MSCs are a heterogeneous population that has been described to have immunoregulatory potential in various immunological cell populations [28], making them candidates for cell therapy in various pathologies associated with inflammation. Several studies have already been implemented on the administration of these cells in patients with acute myocardial infarction, chronic ischemic cardiomyopathy, chronic heart failure and dilated cardiomyopathy, showing improvements in the disease [29]. However, to date, there are no studies that delve into the effect of BM-MSCs on monocyte-derived macrophages from patients with STEMI, who present with a high production of inflammatory cytokines and an inefficient anti-inflammatory response, favoring disease persistence and modulating other immune cell populations [30].

Myocardial infarction triggers an intense inflammatory response that is implicated in the pathogenesis of postinfarction remodeling and heart failure [31]. In our results, we found a higher circulating concentration of the cytokines that characterize M1 polarization in macrophages (IL-1β, IL-6, IL-12p70 and IP-10) in patients with STEMI than in healthy subjects. No differences were observed between groups regarding the characteristic molecules of M2 polarization. Previous evidence, similar to our results, has described an increase in circulating IP-10 in patients with acute myocardial infarction associated with the size of the infarct [32]; other investigations have not found a change in TNF-α between healthy subjects and patients, similar to our results [20]. Consistent with our results, another study demonstrated high concentration levels of IL-1β, IL-6 and IL-12p70 in patients with STEMI [33] and showed that these were associated with a lower improvement response in patients with advanced heart failure [34]. These results indicate a highly inflammatory environment in patients with STEMI, which could favor postinfarction pathogenesis.

Several studies have shown the importance of reparative and remodeling mechanisms in myocardial infarction [31]. In this sense, MSCs are therapeutic candidates due to their ability to modify cellular activity [29]. Initially, we determined the effect of BM-MSCs on macrophage maturation through CD68 expression. We found that BM-MSCs induce an increase in CD68 expression under M1- and M2-polarizing conditions. In this regard, M1 and M2 macrophages express CD68 [35]. Moreover, MSCs induce the expression of CD68 [36], and this marker facilitates the transition from monocytes to macrophages [25]. These results indicate that BM-MSCs favor the generation of both M1 and M2 macrophages from patients with STEMI.

MSCs can regulate the inflammatory response by suppressing leukocytes and triggering anti-inflammatory subsets in innate immunity and adaptive immunity [16,26]. In myocardial infarction, M1-like macrophages predominate; these secrete tumor necrosis factor and interleukin-23 and promote the inflammatory response [37]. We found that M0 macrophages from patients cultured in the presence of BM-MSCs expressed high levels of HLA-DR, CD80 and CD86; the expression of these markers was maintained under M1 conditions in patients with STEMI. Compared with macrophages from healthy subjects, the expression of CD80, CD86 and HLA-DR was decreased [38,39]. This result may be because macrophages with inflammatory characteristics have greater difficulty losing these markers due to an increase in the concentration of nitric oxide [40]. However, M1 macrophages from patients with STEMI cocultured with BM-MSCs increased the expression of M2 markers (CD14, CD163 and CD206). Previous reports have indicated that macrophages polarized toward an M1 phenotype have the potential to acquire an M2 phenotype in response to the presence of IL-4 and IL-13 [38], which are secreted by MSCs [41,42]. Conversely, we demonstrate that under conditions of M2 macrophages derived from patients, BM-MSCs contribute to a greater increase in M2 markers compared to macrophages from subjects with MSCs; the results of previous reports agree with our results [24]. These results indicate that BM-MSCs favor the expression of M2 markers in M1 and M2 macrophages from patients with STEMI, which could modulate the inflammatory response in infarction.

Previous reports have established that MSCs express regulatory genes in coculture with macrophages [43]. In our results, we found that the coculture of BM-MSCs with M1 macrophages increased the presence of IL-10 and IL-1RA, both of which are cytokines related to M2 polarization. In this regard, it has been shown that the presence of MSCs increases the concentration of IL-10 [36], which induces the CCL2/CXCL12 heterodimer, favoring the polarization of M2 macrophages [44]. Another study reported that the administration of MSCs increases the expression of IL-10 in damaged tissue, as well as an anti-inflammatory environment, which resulted in an improved recovery of cardiac function in a murine model of myocardial infarction in diabetic mice [45]. Interestingly, we identified the presence of IL-1Ra under M1 and BM-MSC conditions. Previous data indicate that this cytokine is secreted by inflammatory macrophages [46] and MSCs, supporting an increase in the expression of IL-10 and CD206, similar to our results [47,48]. These results show that there is communication in which BM-MSCs modulate the polarization of macrophages from patients with STEMI toward an M2 phenotype through IL-10 and IL-1Ra.

MSCs exert their immunomodulatory effects through the secretion of molecules such as IL-10 [24]. We found that under basal conditions, BM-MSCs express IL-10 and M-CSF and, to a lesser extent, CD54, which are previously demonstrated characteristics [28,49]. Additionally, we determined that coculture with patient-derived M0 macrophages decreased the expression of IL-10, MCS-F and CD54 in BM-MSCs compared to coculture with macrophages from healthy donors. Under M1 macrophage conditions, the expression of IL-10 and M-CSF in BM-MSCs reached levels similar to those in BM-MSCs without treatment, and increased the expression of CD54 on BM-MSCs under M1 conditions. Additionally, we found that BM-MSCs express IL-10 and M-CSF under M2 macrophage conditions. It has been reported that the IL-10 and M-CSF present in MSCs promote M2 polarization [48,50] and reduce the presence of IL-6 in damaged tissue, reducing myocardial injury in renovascular hypertension in a murine model [49]. Additionally, it has been reported that the presence of M-CSF favors the restoration of cardiac function in a murine model after ischemic injury because it favors VEGF-mediated angiogenesis [51]. Another group reported that the expression of CD54 in MSCs generates a CD54/CD54 interaction with macrophages, favoring the production of IDO and PGE2 [49], and in the presence of M-CSF, favoring the expression of CD206 in macrophages [50].

Previous reports in animal models have established that Tregs are critical in inflammatory regulation in myocardial infarction [27]. We determined that BM-MSCs increase the generation of Tregs with a CD4^+^CD25^+^FoxP3^+^ phenotype independent of the M1 or M2 microenvironment. It has previously been reported in vitro that the presence of macrophages and MSCs increases the generation of Tregs [52]. Furthermore, we found that cocultures of BM-MSCs with M1 or M2 macrophages derived from STEMI patients did not decrease IL-10 and IL-1Ra expression. It has been reported that macrophages previously exposed to MSCs favor the generation of Tregs [42]. The results suggest that BM-MSCs are essential in the generation of Tregs independent of the M1 or M2 microenvironment that prevails in STEMI, which contributes to the regulation of inflammation in infarction. Furthermore, a drastic decrease in Tregs has been reported in STEMI [53].

## 4. Materials and Methods

### 4.1. Experimental Protocol

This study was approved by the Human Ethics and Medical Research Committee of the Instituto Mexicano del Seguro Social (IMSS) on 20 May 2018, and registered (R-2018-785-044). It was conducted according to the Helsinki Declaration guidelines, and all patients provided written informed consent. This study included twenty healthy volunteers who served as a control group of normolipidemic 30- to 50-year-old volunteers without cardiovascular risk factors or clinically apparent atherosclerotic disease. Plasma cytokine levels were measured in 20 healthy subjects, and experimental assays were performed in 6 healthy subjects.

### 4.2. Patient Population

This study included 59 patients whose samples were collected on day 1 of STEMI onset. All patients underwent medical treatment at the Hospital de Cardiología of Centro Médico Nacional Siglo XXI of IMSS, Mexico City. Plasma cytokine levels were measured in 59 patients, and experimental assays were performed in 6 patients. STEMI was diagnosed based on the following three criteria: (1) chest pain > 30 min, with or without shortness of breath, sweating, nausea, and/or vomiting; (2) ST-segment elevation and/or abnormal Q-wave on an electrocardiogram and/or the presence of emerging left bundle branch block; and (3) elevated troponin levels, 10% higher than the 99th percentile of the upper limit of the reference value or elevated creatinine kinase (CK) levels, and/or a myocardial band (MB) fraction higher than the 99th percentile of the upper limit of the reference value. Patients with hemodynamic instability, electrical shock, mechanical complications due to infarction, malignancies, hematological and immunological disorders, any other inflammatory conditions or infections that are likely to be associated with an acute phase response, previous immunosuppressive or anti-inflammatory treatments, serum creatinine ≥ 1.5 mg/dL or known allergic reactions to iodine contrast medium were excluded from this study. Cardiac catheterization was performed in all cases, with successful percutaneous coronary intervention (primary angioplasty and stent placement in the artery responsible for the infarction) being characterized by the presence of TIMI 3 flow after the procedure; this was followed by an evaluation conducted by two independent and blinded observers and using a balloon-door time of 92–140 min. Blood samples were collected via puncture of the antecubital vein before performing coronary angiography during the patients’ stay in the emergency department and from the same subjects 5 days after the procedure during their hospital stay. All patients received optimal anti-ischemic therapy (dual antiplatelet therapy with aspirin and clopidogrel, unfractionated heparin, intravenous nitroglycerine, beta blockers, and statins), calcium channel blockers, and ACE inhibitors, as needed. All patients underwent laboratory tests, including measurements of total cholesterol, triglycerides, high-density lipoprotein cholesterol, low-density lipoprotein cholesterol, white blood cell counts and creatinine.

### 4.3. Isolation and Culture of BM-MSCs

Bone marrow cells were collected according to institutional ethics guidelines, with informed consent obtained from hematologically normal donors from the Mexican Institute for Social Security (IMSS), Mexico City, Mexico. Samples of BM-MSCs (*n* = 6) were obtained from healthy donors. Briefly, BM-MSCs were enriched using a negative selection procedure (RosetteSepTM system; StemCell Technologies Inc. (STI), Vancouver, BC, Canada), and 50 μL of mesenchymal cell enrichment cocktail (monoclonal antibodies against CD3, CD14, CD19, CD38, CD66b and glycophorin A) was added per milliliter of non-diluted BM aspirate. The sample was incubated for 20 min at room temperature and then diluted with PBS containing 2% fetal bovine serum (FBS). Afterward, the diluted sample was layered on top of Ficoll density medium (Pharmacia Biotech, Uppsala, Sweden) and centrifuged for 25 min at 300× *g*. The plasma interface was washed with PBS containing 2% FBS and 1 mM of EDTA. Cells were then maintained in low-glucose Dulbecco’s medium (HyClone). Afterward, the cells were resuspended in low-glucose Dulbecco’s modified Eagle medium (LgDMEM; Gibco BRL, Rockville, MD, USA) supplemented with 10% FBS and seeded at a density of 0.2106 cells/cm^2^ into T25 cell culture flasks (Corning Inc., Costar, New York, NY, USA). After 4 days, the nonadherent cells were removed via pipetting, and fresh medium was added. Every 5 days, the medium was changed. When the cultures reached 80% confluence, they were trypsinized (0.05% trypsin, 0.53 mM EDTA; Gibco BRL, New York, NY, USA) and subcultured at a density of 0.01106 cells/cm^2^ into T75 flasks (Corning). At the fifth passage, cells were used for culture assays. The phenotypes of the BM-MSCs were characterized through the analysis of the molecules CD90, CD73, CD13, CD34, CD31, CD14, CD105, CD29, HLA-DR and CD10 (BD Biosciences, San Diego, CA, USA). The functional capacity of differentiation to the osteogenic and proliferative capacity of BM-MSCs was analyzed as previously described [54].

### 4.4. Monocyte Isolation

Peripheral blood mononuclear cells (PBMCs) were collected from healthy volunteers and STEMI patients via density centrifugation using Lymphoprep (Axis-Shield, Oslo, Norway). Briefly, blood samples were mixed with an equal volume of phosphate-buffered saline (PBS), pH 7.4, layered over 3 mL of Lymphoprep, and centrifuged at 700× *g* for 30 min. The recovered PBMCs were washed three times with PBS (pH 7.4). The PBMCs were frozen at −80 °C until use. Monocytes were then isolated from the PBMCs previously frozen via negative selection. Briefly, PBMCs were incubated with FcR blocking reagent and a cocktail of biotin-conjugated monoclonal anti-human antibodies against antigens that are not expressed on human monocytes. Magnetic microbeads were coupled to anti-hapten monoclonal antibodies and depleted using a magnetic column (Classical Monocyte Isolation Kit, Miltenyi Biotec, Bergisch Gladbach, Germany). Monocytes were maintained in RPMI-1640 medium (HyClone, Marlborough, MA, USA) supplemented with 10% fetal bovine serum (FBS) (Gibco BRL, Rockville, MD, USA), 2 mM of l-glutamine (Biowest), 1X penicillin/streptomycin (Biowest) and 100 µg/mL of gentamicin (Biowest, Nuaillé, PC, France).

### 4.5. Macrophage Polarization

Monocytes were polarized toward M1 and M2 macrophage phenotypes [55]. Monocytes were cultured in RPMI-1640 medium (HyClone)/low-glucose DMEM (HyClone) (45%/45%) and 10% FBS (Gibco) in the presence of inducer medium 1 (M1) that contained granulocyte–macrophage colony-stimulating factor (GM-CSF) (50 ng/mL) (Miltenyi Biotec) for 96 h. Then, the culture medium was replaced with fresh culture medium containing lipopolysaccharides (LPS) (100 ng/mL) (Sigma–Aldrich, St. Louis, MO, USA) and interferon gamma (IFNγ) (50 ng/mL) (Miltenyi Biotec), and the cells were incubated for 48 h. To induce M2 polarization, monocytes were cultured in RPMI-1640 medium (HyClone)/low-glucose DMEM (HyClone) (45%/45%) and 10% FBS (Gibco) in the presence of inducer medium 2 (M2) that contained macrophage colony-stimulating factor (M-CSF) (50 ng/mL) (Miltenyi Biotec) for 96 h. Then, the medium was removed, and new medium containing interleukin 4 (IL-4) (Miltenyi Biotec) (50 ng/mL) and interleukin 13 (IL-13) (40 ng/mL) (Miltenyi Biotec) was added, and the cells were incubated for 48 h. Negative control monocytes were cultured with only culture medium in the absence of cytokines (M0).

### 4.6. Macrophage/BM-MSC Cocultures

Monocytes were cocultured for 6 days in the presence or absence of BM-MSCs at a ratio of 5:1 (monocytes:BM-MSCs) in a system with cellular contact or a Transwell system with a pore size of 0.4 μm (Corning, Corning, NY, USA). The cocultures were maintained in the absence of MSCs (control, M0) or in the presence of M1 or M2 polarization inducer medium. Macrophages cultured in the absence of MSCs were used as controls to perform the fold change.

### 4.7. Cytokine Assays

To determine the concentrations of the soluble molecules present in the plasma and supernatants of cocultures of BM-MSCs/macrophages, and of the macrophages alone (control) incubated in M0, M1 or M2 inducer medium, the plasma and supernatants were stored at −70 °C until use. Cytokine identification was performed using LEGENDplex cytometry beads (BioLegend, San Diego, CA, USA). The kit was used in accordance with the manufacturer’s instructions. Samples were analyzed the same day at a low acquisition rate on an Aurora spectral flow cytometer (Cytek Biosciences, Fremont, CA, USA). The data were analyzed with LEGENDplex software version 9 (BioLegend).

### 4.8. Surface Markers on Macrophages

The macrophages were washed with PBS, stained with 25 μL of the viability marker Ghost Dye Red 780 (TONBO Biosciences, San Diego, CA, USA) (1/1500) for 15 min at room temperature, washed with PBS and blocked with FBS (Corning) at 4 °C for 10 min. On the one hand, we analyzed the maturation of macrophages through CD68 expression on macrophages using anti-human CD68-PE-Cy7 (BioLegend). On the other hand, the characteristic markers of M1 macrophages were determined using the following antibodies: anti-human CD86-PE (BioLegend), CD80-BV421 (BioLegend) and HLA-DR-PE/Cyanine7 (BioLegend). The characterization of M2 macrophages was performed using the following antibodies: anti-human CD14-BV510 (BioLegend), CD163-APC (BioLegend), CD206- and PE-Cy™5 (BD Biosciences). After adding the antibodies, the cells were incubated at 4 °C for 20 min. The cells were then fixed with 1% paraformaldehyde for 10 min at 4 °C and washed with PBS. Acquisitions were performed on an Aurora spectral flow cytometer (Cytek Biosciences), and flow cytometric analysis was performed using FlowJo (Ashland, OR, USA, V10 software) (Appendix A). Macrophages grown in the absence of BM-MSCs and under different polarization conditions were used as controls. The calculation of M1 and M2 polarization was obtained via individual experiments, which were performed to determine the fold increase (compared to the control) in the mean fluorescence intensity of the positive expression of membrane molecules characteristic of M1 or M2 polarization in macrophages cocultured in cell contact for 6 days in vitro.

### 4.9. Intracellular Marker Evaluation

After 6 days of direct coculture between BM-MSCs and macrophages, the cocultures were treated with Golgi Stop reagent (BD Biosciences) to inhibit protein transport and evaluate intracellular molecules. After 12 h of treatment, the cocultures were washed with 1X PBS, stained with 25 μL of the viability marker Ghost Dye Red 780 (TONBO biosciences, San Diego, CA, USA; 1 μL of the stock diluted in 1.499 μL of PBS) for 15 min at room temperature, washed with 1X PBS and blocked with FBS (Biowest) at 4 °C for 10 min. After blocking, the macrophages were identified with anti-human CD45-FITC and anti-human CD68-PECy7 (BD Biosciences) (CD45+CD68+), whereas BM-MSCs were identified with anti-human CD45-FITC (BD Biosciences), anti-human CD90-PE-Cy™5 (BD Biosciences), and anti-human CD54-Pacific Blue (Biolegend) (CD45-CD90+CD54+), followed by incubation at 4 °C for 20 min and washing with 1X PBS (Biowest). The cell membrane was permeabilized following the manufacturer’s instructions (Invitrogen, Carlsbad, CA, USA) to evaluate MSCs, and anti-human IL-10-BV421 (Biolegend) and anti-human M-CSF-PE (R and D Systems) were added to the cells. Acquisitions were made on a spectral flow cytometer Aurora (Cytek Biosciences) and analyzed with FlowJo V10 software (Appendix A).

### 4.10. Generation of Regulatory T Cells

To assess whether macrophages in the presence of BM-MSCs generated Tregs, macrophages were first treated with BM-MSCs without cellular contact in a 0.4 μm Transwell system (Corning) for 6 days in the presence of M0, M1 or M2 inducer medium, after which they were cocultured with CD4^+^ T cells selected using CD4 MicroBeads (Miltenyi Biotec) in a 1:1 ratio (macrophages:T cells) in RPMI medium (HyClone) containing 10% FBS (Corning) in the presence of anti-CD2/CD3/CD28 beads (1:1 T cell ratio) (Miltenyi Biotec). After 5 days of coculture, T lymphocytes were harvested and blocked with FBS (Biowest) at 4 °C for 15 min. After blocking, anti-human CD4-PE (BD Biosciences) and anti-human CD25-FITC (BD Biosciences) antibodies were added. The cells were then permeabilized following the manufacturer’s instructions with the FoxP3 staining buffer set kit (Invitrogen). After permeabilization, anti-human FoxP3-APC (Biolegend) was added. Acquisitions were made on a spectral flow cytometer Aurora (Cytek Biosciences) and analyzed with FlowJo V10 software (Appendix A). CD4+ T lymphocytes cultured in the absence of macrophages were used as a control.

### 4.11. Statistical Analysis

All statistical analyses were performed using GraphPad Prism version 9 (GraphPad Software, Inc., San Diego, CA, USA), and a level of *p* < 0.05 was considered statistically significant. The levels of circulating cytokines were analyzed using the Mann–Whitney test and Kruskal–Wallis test to evaluate cocultured cells. The experimental study included six independent experiments and measured circulating cytokine levels in fifty-nine patients and twenty healthy subjects.

## 5. Conclusions

Our study demonstrates that inflammatory conditions prevail in patients with STEMI, and M1 macrophages play an essential role in directing this response. In this sense, M1 macrophages from STEMI patients did not show a reduction in M1-associated markers when cultured in contact with BM-MSCs; however, these macrophages increased the expression of markers associated with an anti-inflammatory macrophage and are capable of inducing Tregs via a microenvironment in which IL-10 and IL-1Ra prevail. Additionally, BM-MSCs enhance the expression of cytokines, Treg generation and M2 markers in M2 macrophages from patients with STEMI. The importance of these discoveries lies in the fact that BM-MSCs attenuate the inflammatory activity of M1 macrophages and enhance the activity of M2 macrophages, which could improve cardiac tissue damage in patients with STEMI.

## Figures and Tables

**Figure 1 ijms-24-16257-f001:**
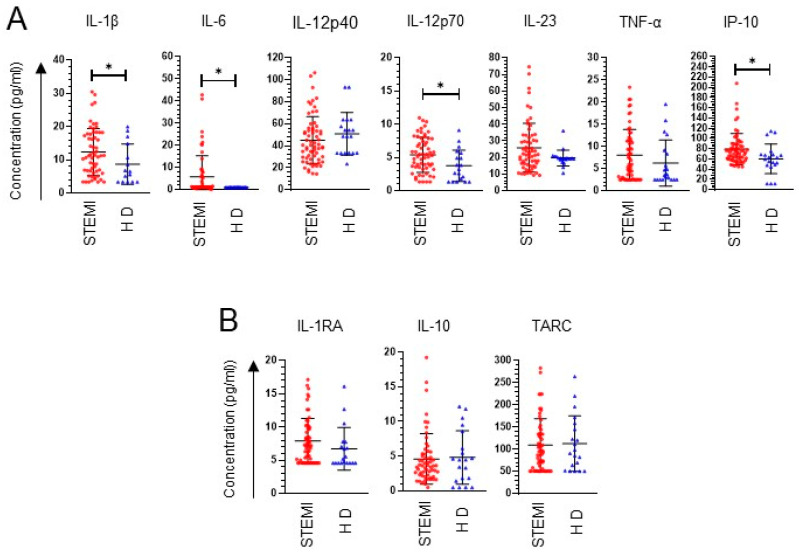
Circulating pro- and anti-inflammatory cytokines in patients with STEMI. (**A**) Dot plot of the concentration of molecules characteristic of M1 polarization in plasma. (**B**) Dot plot of the concentration of molecules characteristic of M2 polarization in plasma. Cytokine levels were determined using flow cytometry. Patients with STEMI (*n* = 59). Healthy donor (HD) (*n* = 20). * Significant difference (*p* < 0.05).

**Figure 2 ijms-24-16257-f002:**
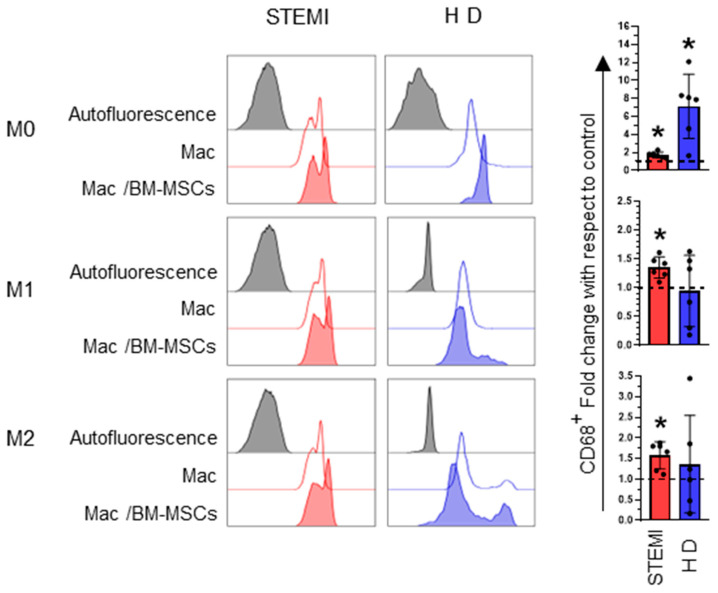
BM-MSCs induce the maturation of macrophages from patients with STEMI. Representative histograms of CD68 expression on macrophages and dot plots of the mean, with standard deviation of the fold increase in positive expression relative to the control. * Significant difference (*p* < 0.05); *n* = 6. M0: absence of inducer medium; M1: inducer medium for M1 polarization; M2: inducer medium for M2 polarization. Dotted lines indicate macrophages behavior in the absence of BM-MSCs and under different polarization conditions were used as controls. * Difference between macSTEMI-BM-MSCs or macHD-BM-MSCs cocultures versus macrophages only. The black dot on the bar graph indicates data from an assay. Patients with STEMI. Healthy donor (HD). Mac: macrophages only in culture medium. BM-MSCs: bone marrow mesenchymal stem/stromal cells.

**Figure 3 ijms-24-16257-f003:**
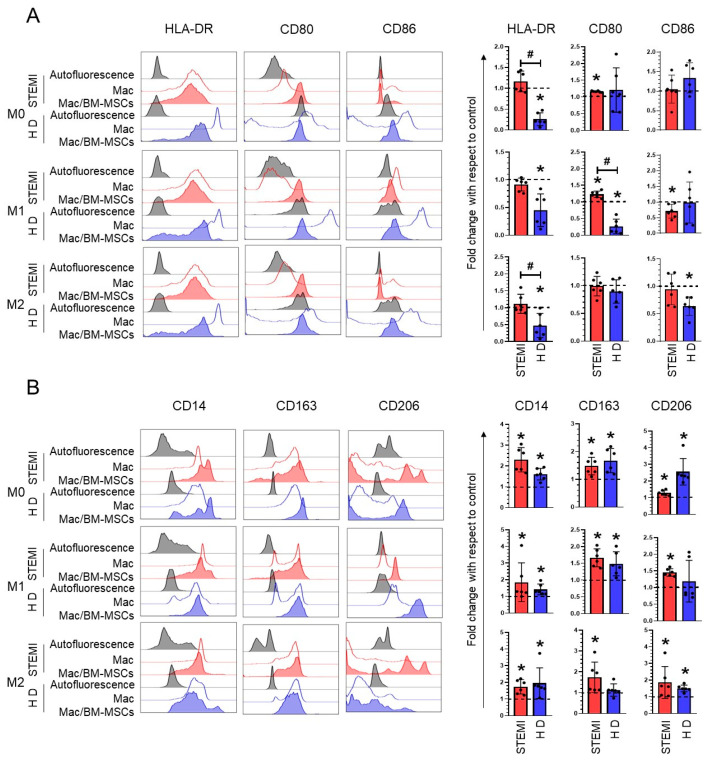
The coculture of macrophages derived from STEMI patients with BM-MSCs increased the expression of M2 markers in macrophages. (**A**) Histograms and dot plots of the mean with standard deviation of the fold increase in the positive expressio of characteristic markers of M1 polarization with respect to the control. (**B**) Histograms and dot plots of the mean with standard deviation of the fold increase in the positive expression of characteristic markers of M2 polarization with respect to the control. Significant difference with respect to the control (*p* < 0.05); *n* = 6. M0: absence of inducer medium; M1: inducer medium for M1 polarization; M2: inducer medium for M2 polarization. Patients with STEMI. Dotted lines indicate macrophages behavior in the absence of BM-MSCs and under different polarization conditions were used as controls. * Difference between macSTEMI-BM-MSCs or macHD-BM-MSCs cocultures versus macrophages only. # Difference between the macSTEMI-BM-MSCs and macHD-BM-MSCs coculture groups. Healthy donor (HD). Mac: macrophages only in culture medium. BM-MSCs: bone marrow mesenchymal stem/stromal cells.

**Figure 4 ijms-24-16257-f004:**
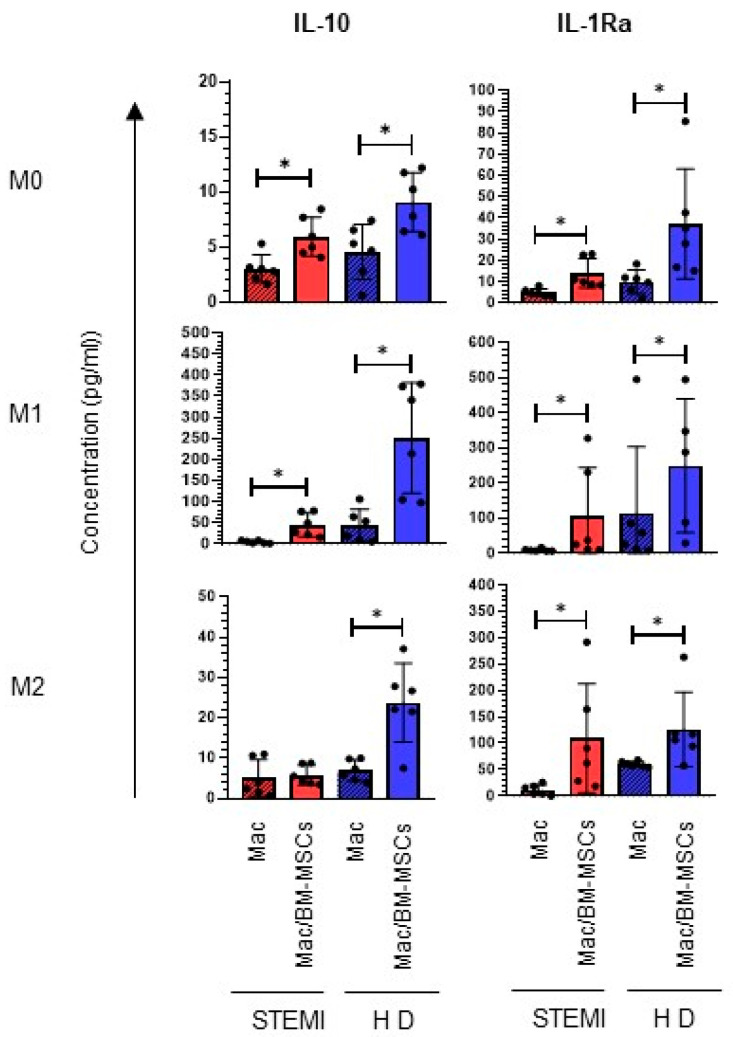
Coculture of BM-MSCs with macrophages derived from patients with STEMI increases anti-inflammatory cytokines. Determination of IL-10 and IL-1Ra levels in cocultures of BM-MSCs with macrophages under different conditions. The graphs show the mean with standard deviation of the concentration (pg/mL) in the supernatant of molecules characteristic of M2 polarization. Significant difference with respect to the control (*p* < 0.05); *n* = 6. M0: absence of inducer medium; M1: inducer medium for M1 polarization; M2: inducer medium for M2 polarization. * Difference between groups. The black dot on the bar graph indicates data from an assay. Patients with STEMI. Healthy donor (HD). Mac: macrophage only. BM-MSCs: bone marrow mesenchymal stem/stromal cells.

**Figure 5 ijms-24-16257-f005:**
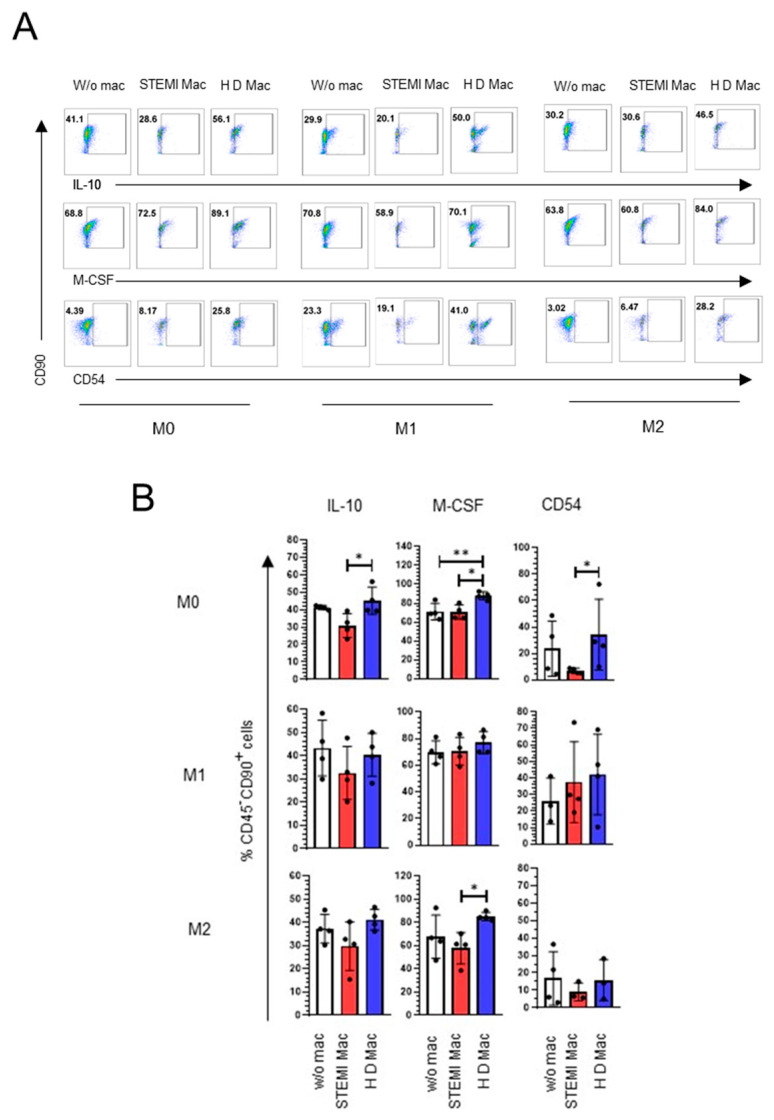
Effect of macrophages on the expression of modulating molecules in BM-MSCs. Expression of IL-10, M-CSF and CD54 in BM-MSCs cocultured with macrophages. (**A**) Representative dot plot of flow cytometry in BM-MSCs exposed to M0, M1 and M2 macrophages. (**B**) Graph representing the changes in the expression of IL-10, M-CSF and CD54 in BM-MSCs. A significant difference was considered with *p* < 0.05; *n* = 6. M0: absence of inducer medium; M1: inducer medium for M1 polarization; M2: inducer medium for M2 polarization. * Difference between BM-MSCs cocultured with STEMI mac and HD mac. ** Difference between BM-MSCs w/o mac and cocultured with HD Mac. The black dot on the bar graph indicates data from an assay. Patients with STEMI. Healthy donor (HD). W/o mac: without macrophages. BM-MSCs: bone marrow mesenchymal stem/stromal cells.

**Figure 6 ijms-24-16257-f006:**
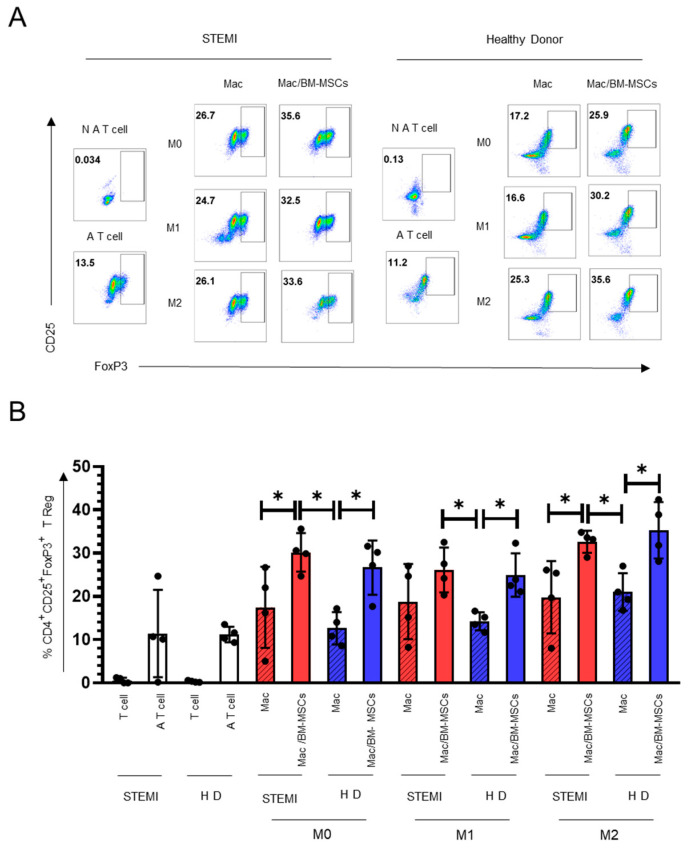
Coculture with BM-MSCs increases the capacity of macrophages derived from patients with STEMI to generate regulatory T cells. (**A**) Dot plots representative of the percentage of CD4^+^CD25^+^FoxP3^+^ regulatory T cells. (**B**) Graph with the standard deviation of the mean of the percentage of CD4^+^CD25^+^FoxP3^+^ regulatory T cells. * Significant difference (*p* < 0.05) between percentage of CD4^+^CD25^+^FoxP3^+^ cells (STEMI or HD) cocultured with BM-MSCs primed macrophages (STEMI or HD); *n* = 6. M0: absence of inducer medium; M1: inducer medium for M1 polarization; M2: inducer medium for M2 polarization. The black dot on the bar graph indicates data from an assay. Patients with STEMI. Healthy donor (HD). Mac: macrophages only in culture medium. BM-MSCs: bone marrow mesenchymal stem/stromal cells. T cell: T cell only with culture medium. A T cells: activated T cells.

**Table 1 ijms-24-16257-t001:** Characteristics of the study population. The values are expressed as the mean ± SD.

	Healthy Subjects *n* = 20	STEMI Patients *n* = 59
Demographics		
Age, years, mean ± SD	34 ± 11	66 ± 12
Female, *n* (%)	11 (55%)	4 (7%)
Male *n* (%)	9 (45%)	55 (93%)
Obesity, *n* (%)	0	28 (47%)
Smoking, *n* (%)	0	33 (56%)
Diabetes mellitus, *n* (%)	0	27 (45%)
Hypertension, *n* (%)	0	37 (62%)
Hyperlipidemia, *n* (%)	0	37 (62%)

## Data Availability

The data presented in this study are available in the article.

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
