# Peer review of "Human Bone Marrow Mesenchymal Stem Cells Promote the M2 Phenotype in Macrophages Derived from STEMI Patients"

_ijms, 2023, doi:10.3390/ijms242216257_

Round 1

Reviewer 1 Report

Comments and Suggestions for Authors

Thanks for doing this interesting study.

Regarding the control group, were you able to investigate healthy donors "with and without comorbidities" of the same age as the STEMI?  I wonder if aging will have a difference if any.

Also, did you find differences between males and females in each group? 

Comments on the Quality of English Language

The Y axis of the PCR graphs in figures 2 & 3 is labeled fold chance, is that a correct label or it should be fold change?

Author Response

We appreciate the suggestions made.

Regarding the control group, were you able to investigate healthy donors "with and without comorbidities" of the same age as the STEMI?  I wonder if aging will have a difference if any.

The healthy donors included in the study had no comorbidities. On the other hand, it is difficult to age-match healthy donors with patients because age is considered an independent risk factor in the disease.

Also, did you find differences between males and females in each group?

We did not find differences between men and women in the groups

Comments on the Quality of English Language

The Y axis of the PCR graphs in figures 2 & 3 is labeled fold chance, is that a correct label or it should be fold change?

We have made corrected the figures.

Reviewer 2 Report

Comments and Suggestions for Authors

The whole paper was good, but I think some things must be rewritten or reviewed.

Introduction section must be extended as it doesn't provide enough background to the topic that is been presented.

Second, materials and methods section, can be improved giving more data on how everything was done.

Finally, I prefer the normal paper format: Introduction, M&M, Results, Dsicussion, Conclusion; as it is easier to follow the whole process, just personal opinion.

Author Response

We appreciate the suggestions made.

Introduction section must be extended as it doesn't provide enough background to the topic that is been presented.

We appreciate the suggestion. In this new version, we have made changes to the introduction to make it clearer.

Second, materials and methods section, can be improved giving more data on how everything was done.

In this version, we have expanded the materials and methods section.

Finally, I prefer the normal paper format: Introduction, M&M, Results, Dsicussion, Conclusion; as it is easier to follow the whole process, just personal opinion.

We appreciate the suggestion. However, we have followed the journal format.

Reviewer 3 Report

Comments and Suggestions for Authors

The study is a well-written and clearly presented manuscript that seeks to characterize the functional impact of Human bone marrow mesenchymal stem cells on M2 phenotype in macrophages derived from STEMI patients.

 An observation made by the authors is that BM-MSCs can modulate the M1 macrophage 34 response, which could improve cardiac tissue damage in patients with STEMI. However, below are my comments to improve the manuscript.

Major comments 

  1. The introduction can be expanded.
  2. Did authors perform experiments where they checked mRNA expressions in addition to the protein expressions for the co-cultured in vitro experimental models?
  3. As only histograms are represented in the case of performed flow cytometry experiments, the gating strategy is not clear. Authors can include their gating strategy at least in the supplementary material for a better understanding
  4. In addition to histograms/ dot plots, it would be more informative to include representative MFI values for the performed flow cytometry measurements.

It would be helpful if authors summarized the percentage of single 

  1. and double positive population percentages of dot plots in Figures 5 and 6.

Minor comments

  1. The text font in some parts of the materials and methods part is not consistent (section 4.3)

Author Response

We appreciate the suggestions made.

Major comments

  1. The introduction can be expanded.

In this new version you find the introduction expanded.

  1. Did authors perform experiments where they checked mRNA expressions in addition to the protein expressions for the co-cultured in vitro experimental models?

In this study we did not determine the expression of mRNA in the tests carried out, we will take the suggestion into account for future studies, thank you.

  1. As only histograms are represented in the case of performed flow cytometry experiments, the gating strategy is not clear. Authors can include their gating strategy at least in the supplementary material for a better understanding.

We appreciate the suggestion. We made a section of supplementary material that describes the strategy to follow in flow cytometry assays.

  1. In addition to histograms/ dot plots, it would be more informative to include representative MFI values for the performed flow cytometry measurements.

We carried out MFI analyses of the different assays and found that the biological phenomenon was maintained in all of them. However, the data did not have high uniformity. Therefore, we decided to determine changes in expression.

It would be helpful if authors summarized the percentage of single and double positive population percentages of dot plots in Figures 5 and 6.

We appreciate the suggestion. In Figure 5, the data shown are percent simple due to methodological limitations in evaluating the three molecules at the same time. Meanwhile, in Figure 6 we analyze the expression of CD4+CD25+FoxP3+ in a single cell and this is how they are shown in the figure.

Minor comments

  1. The text font in some parts of the materials and methods part is not consistent (section 4.3)

We standardized the font size in section 4.3.

Round 2

Reviewer 3 Report

Comments and Suggestions for Authors

Thank you. I guess the manuscript looks fine.